# Smartphone-Based Body Location-Independent Functional Mobility Analysis in Patients with Parkinson’s Disease: A Step towards Precise Medicine

**DOI:** 10.3390/jpm12050826

**Published:** 2022-05-19

**Authors:** Diogo Vila-Viçosa, Mariana Leitão, Raquel Bouça-Machado, Filipa Pona-Ferreira, Sara Alberto, Joaquim J. Ferreira, Ricardo Matias

**Affiliations:** 1Kinetikos, 3030-199 Coimbra, Portugal; dvicosa@kinetikoshealth.com (D.V.-V.); salberto@kinetikoshealth.com (S.A.); 2CNS—Campus Neurológico Sénior, 2560-280 Torres Vedras, Portugal; marianaleitao.ft@gmail.com (M.L.); raquelbouca@cnscampus.com (R.B.-M.); filipaponaferreira@campus.ul.pt (F.P.-F.); joaquimjferreira@cnscampus.com (J.J.F.); 3Instituto de Medicina Molecular João Lobo Antunes, 1649-028 Lisbon, Portugal; 4Laboratory of Clinical Pharmacology and Therapeutics, Faculdade de Medicina, Universidade de Lisboa, 1649-028 Lisbon, Portugal; 5Physics Department & Institute of Biophysics and Biomedical Engineering (IBEB), Faculty of Sciences, University of Lisbon, Campo Grande, 1749-016 Lisbon, Portugal

**Keywords:** Parkinson’s disease, digital health, inertial sensors, wearable technology, gait evaluation, spatiotemporal gait metrics

## Abstract

Ecological evaluation of gait using mobile technologies provides crucial information regarding the evolution of symptoms in Parkinson’s disease (PD). However, the reliability and validity of such information may be influenced by the smartphone’s location on the body. This study analyzed how the smartphone location affects the assessment of PD patients’ gait in a free-living environment. Twenty PD patients (mean ± SD age, 64.3 ± 10.6 years; 9 women (45%) performed 3 trials of a 250 m outdoor walk using smartphones in 5 different body locations (pants pocket, belt, hand, shirt pocket, and a shoulder bag). A method to derive gait-related metrics from smartphone sensors is presented, and its reliability is evaluated between different trials as well as its concurrent validity against optoelectronic and smartphone criteria. Excellent relative reliability was found with all intraclass correlation coefficient values above or equal to 0.85. High absolute reliability was observed in 21 out of 30 comparisons. Bland-Altman analysis revealed a high level of agreement (LoA between 4.4 and 17.5%), supporting the use of the presented method. This study advances the use of mobile technology to accurately and reliably quantify gait-related metrics from PD patients in free-living walking regardless of the smartphone’s location on the body.

## 1. Introduction

From the early stages of disease, PD patients experience impairments in functional mobility, i.e., the ability to move independently and safely in order to accomplish activities of daily living [1,2]. Indeed, mobility, and particularly walking, is often rated by people living with chronic conditions as one of the most important clinical outcomes, its loss becoming a major cause of disability and diminished independence [3,4,5].

Current assessments focus on three dimensions of mobility: capacity, perception and performance. In PD, capacity and perception are evaluated clinically, through the Movement Disorders Society Unified PD Rating Scale (MDS-UPDRS) parts III and II, respectively, while no standardized way of measuring mobility performance exists currently [4,6]. Thus, while accurate quantitative information regarding the mechanics of PD patients’ gait is among the most promising outcomes that enable early diagnosis and assessment of disease progression and therapeutic interventions, mobility measurements are obtained mainly through subjective approaches and only infrequently under supervised conditions. Such methods are subject to inter-rater variability, the Hawthorne effect and recall and other biases, bringing into question the accuracy and ecological validity of the results [3,7].

The emergence of and advances in technology-based objective measures of PD gait provide the opportunity to broaden the scope of functional mobility assessments to quantitatively and reliably capture relevant, ecologically valid outcomes in a continuous and unobtrusive manner [2,4,5,8,9]. However, it has been demonstrated that remote mobility measurement systems requiring the usage of a wearable sensor can have dropout rates of up to 32% after 6 months and 50% after a year [10]. Smartphones, ubiquitous, portable, user-friendly and affordable devices with one or several embedded inertial sensors, offer a promising new avenue for the analysis of the mobility performance of PD patients outside clinical environments, complementing the capacity measures obtained in-consultation and yielding rater-independent, quantitative outcomes with higher ecological validity [11,12]. The validity of smartphone applications for this purpose has been shown in healthy subjects and in PD patients, albeit mostly in supervised environments, requiring the device to be worn in a predetermined location and orientation, and not accounting for the heterogeneity of smartphone usage among the population [13,14,15]. These factors contribute to reduced ecological validity and patient compliance, decreasing the quantity and quality of recorded data and bringing into question the reliability and validity of the measurements obtained through those algorithms in an unsupervised environment. In fact, 26% of PD symptom-monitoring applications are used only once, and 74% are not used more than 10 times, emphasizing the importance of creating no-burden solutions to passively monitoring mobility [10].

Reliability and validity of smartphone-based gait analysis with the device worn in different body locations have been demonstrated in healthy subjects, though solely under standardized conditions [16,17]. Furthermore, algorithms trained using data from healthy subjects may not perform properly when applied to PD patients due to the different spatiotemporal gait dynamics between the two groups [13,18]. Comparison between different studies is also limited due to the high heterogeneity between gait analysis protocols employed.

It is apparent that there is a necessity to measure relevant mobility outcomes in PD patients continuously and objectively to provide more personalized and ecological perspectives on patients’ real-world performance [19]. In addition, the technology employed for this purpose should not present a burden for the user, and algorithms used to calculate mobility parameters should be validated in PD patients, to the highest degree possible, under ecologically valid conditions. This study aims to assess the effects of carrying smartphones on different body locations when determining gait spatiotemporal measures of patients with PD during outdoor walking.

## 2. Materials and Methods

### 2.1. Study Design

A cross-sectional clinical study was conducted.

### 2.2. Objective

To analyze how the smartphone location affects the assessment of Parkinson’s disease patients’ gait in a free-living environment.

### 2.3. Participants

Twenty participants (11 male) were recruited from CNS—Campus Neurológico Sénior, a tertiary specialized movement disorders center in Portugal. Patients were eligible if they had a diagnosis of probable or clinically established PD (according to the International Parkinson and Movement Disorder Society criteria). Exclusion criteria were the inability to adopt a standing position and/or to walk independently; a history of falls (one fall in the previous month); the inability to correctly respond to the assessment protocol according to the clinician’s judgment; or the presence of a cardiovascular, pulmonary, or musculoskeletal condition that according to the clinician’s judgment would affect a patient’s ability to participate in the study. The study was undertaken with the understanding and written consent of each participant, with approval from the CNS Ethics Committee (Ref. 5-2021), and in compliance with national legislation and the Declaration of Helsinki. Participants were required to agree to all aspects of the study and were able to leave the study at any time.

### 2.4. Clinical Assessment Protocol

Patients were assessed in ON-state medication by a physiotherapist who specialized in movement disorders. The following parameters were collected:Demographic and clinical data;Disease severity: Movement Disorder Society—Unified Parkinson’s Disease Rating Scale (MDS-UPDRS) total score and score from each subsection [20] and Hoehn and Yard scale [21] Clinical and Patient Global Impression (CGI and PGI, respectively) of Severity [22];Functional mobility: The Timed Up and Go (TUG) test;Quality of Life: the Parkinson’s Disease Quality of Life Questionnaire (PDQ-39).

### 2.5. Gait Assessment Protocol

Subjects were invited to walk three times around the CNS outdoor garden at self-selected speed while in clinical “on” state medication (a video and a picture of the entire path are available as Appendix A)—corresponding to approximately 3 × 250 m—with a rest period of 5 min between trials. The 250 m route included level parts, ramps (going both up and down), different terrains, and several sharp changes in direction (both to the left and to the right). The patients were allowed to gesture and speak with the accompanying health care professional. Each participant carried four smartphones: one on the right pocket of their pants (Pants), one on a belt bag (Belt) and another on the hand (Hand). The 4th smartphone was carried in a shirt pocket (Pocket) for ten subjects and in a shoulder bag (Bag) for the other ten (see an illustrative representation in the Appendix A).

### 2.6. Data Collection and Analysis

Outdoor gait-related measurements were obtained using Samsung A20 smartphone inertial sensors—accelerometer, gyroscope and magnetometer—at 100 Hz with mKinetikos™ (Kinetikos, Coimbra, Portugal) application [23] version 2.3.3. To obtain a clear location-independent identification of gait events in this updated version of mKinetikos™—toe-offs and heel-strikes—a protocol with three main steps was developed: sensor orientation, event detection and outlier exclusion (see the pipeline of data processing in the Appendix A). A Madgwik gradient descent IMU orientation estimation was performed to retrieve the vertical component of the acceleration [24]. After this step, an initial event detection was performed where the toe-offs corresponded to peaks in the acceleration signal whereas heel-strikes corresponded to peaks on its first derivative. From an initial list of toe-offs, every pair of toe-offs that was significantly lower—30%—than the previous and the next toe-off was removed. The event detection step was finished by adding virtual events—both toe-offs and heel-strikes—in the cases where a heel-strike was found without a corresponding toe-off. Finally, two outlier rejection steps were performed. The first used the vertical component of the acceleration of each stride—normalized to 100 points—and the first four components of a pretrained principal component analysis (PCA) model. These four components fed a pretrained unsupervised isolation forest model to remove outliers (20%). These two models were trained using all strides collected in this study. The interquartile range rule was then applied to each gait metric to remove values that were larger (lower) than 1.5 × IQR + 3rd (−1st) quartile, where IQR stands for the interquartile range. After the two main steps of this protocol—event detection and outlier exclusion—an average of 61.5% of the initial detected events was maintained (Table 1). From the list of inlier events, gait cycles were constructed as a sequence of an initial heel-strike, an ipsilateral toe-off and the final ipsilateral heel-strike, considering that consecutive heel-strikes correspond to opposite sides. However, using a single smartphone, it was not possible to assign a side for each event. Consequently, to avoid misinterpretations, all presented metrics were calculated for strides instead of steps. Gait parameters were calculated as follows:stride time as the time difference between the final and initial heel strikes of a gait cycle;stance phase duration as the time between a heel-strike event and the following toe-off event;swing phase duration as the time between a toe-off event and the following heel-strike event;stride length: the linear relation of stride length, stride frequency (f) and acceleration variance (v) given by stride length = 0.998 × f + 0.032 × v + 0.798 [23,25];stride velocity as stride length divided by stride time.

### 2.7. Statistical Analysis

Relative and absolute reliability were assessed as follows: the former used the intraclass correlation coefficient (ICC) with a two-way mixed-effects model (absolute agreement) and a confidence interval of 95%. The ICC was interpreted according to Shrout and Fleiss [26] where ICC ≥ 0.75 indicates excellent repeatability, ICC 0.4–0.74 indicates fair-to-high repeatability and ICC ≤ 0.39 indicates poor repeatability; the latter was calculated by computing the standard error of measurement (SEM) according to
SEM = SD × √(1 − ICC)
where SD refers to the standard deviation. SEM was used to provide information on variability over repeated trials, and it was considered excellent if SEM < intra-subject variability (SD between trials) in all spatiotemporal measurements. Minimal detectable change scores (MDC) were also calculated to gain further insight into the minimal amount of change that the computed walk-related measurements had to show to be greater than clinically relevant differences:MDC = SEM × 1.65 × √2.

The 1.65 represents the z-score at the 95% confidence level. The product of SEM multiplied by 1.65 is multiplied by the square root of 2 to account for errors associated with repeated measurements.

Power sample calculation revealed that 20 patients would allow an 80% power to detect a reliability higher than 0.9 with a minimum acceptable reliability of 0.7, assuming a type I error of 5% and a drop out of 10%.

Concurrent validity was evaluated in two steps using Bland-Altman analysis and Spearman’s correlation coefficient (according to Ref. [27], 0.5 indicates high correlation, 0.3 is moderate and 0.1 is small): (i) in a first step by analyzing the agreement between an updated version of mKinetikos™ and the optoelectronic criterion using a data set described in [23]; (ii) and in a second step by quantifying the agreement between the Pants smartphone location—used as the criterion—and each of the remaining locations as described in the gait assessment protocol section. Bias and limits of agreement were calculated using the median and the 2.5% and 97.5% quantiles of the differences between two smartphone locations. Biases were compared with zero using a one-sample Wilcoxon test.

## 3. Results

### 3.1. Demographic and Clinical Data

Demographic and clinical data were collected for the cohort of 20 patients (Table 2). Briefly, 55% (*n* = 11) of the participants were male with a mean age of 64.3 ± 10.6 years. The average Hoehn and Yahr score was 2 ± 0.6, with a mean MDS-UPDRS total score of 48.7 ± 26.5 and a TUG mean of 9.4 ± 3.0.

### 3.2. Reliability

Relative reliability was calculated for each smartphone location: Belt, pants front pocket (Pants), hand, shirt pocket (Pocket) and shoulder bag (Bag). All calculated spatiotemporal gait metrics showed ICC values above or equal to 0.85 (Figure 1 and Table 3).

Absolute reliability was evaluated by computing the SEM and the MDC for each metric. SEM was compared with the corresponding intra-subject variability (Figure 2 and Table 4). For 21 out of 30 comparisons, SEM was lower than the intra-subject variability. The only positive differences occurred for the swing time, stride length, stride cadence and stride duration in the Pants pocket.

### 3.3. Concurrent Validity

The agreement between the presented method and the optoelectronic criterion via Bland-Altman analysis showed median absolute biases of 0.006, 0.005 and 0.000 for temporal metrics, spatial metrics and cadence, respectively, and limits of agreement (LoA) with median absolute values of 0.075 (11.9%), 0.097 (11.3%) and 4.263 (7.5%) (Table 5 and Figure 3).

Excellent agreement was found when comparing the Pants smartphone with the remaining smartphone locations (Table 6 and Appendix A) with median absolute biases of 0.002, 0.006 and 0.111 for temporal metrics, spatial metrics and cadence, respectively (*p* > 0.01 for all comparisons). Additionally, the median absolute values for the LoAs were 0.074, 0.055 and 4.951 (10.3, 7.2 and 9.2%). Very low LoAs were observed for the stride length (median of 0.045 and <8%).

Spearman correlation coefficient (⍴) values were calculated for each comparison with the Pants pocket location (Figure 4). All Spearman correlation coefficient (⍴) values were high (above 0.50; *p* < 0.001). Lower correlations were observed for Hand and for swing time. Stride time and velocity showed very high correlations for all locations (⍴ > 0.84).

## 4. Discussion

In this work, we observed a very strong agreement between spatiotemporal gait metrics measured with smartphones in different body locations, and the measured values showed nonsignificant variations between different trials. Current mobility measures (e.g., walking) include patients’ perspectives on their mobility (patient-reported outcomes) and clinician-reported outcomes about patients’ capacity to walk. Smartphone-based quantification of real-world walking offers a promising strategy for continuous gait analysis outside the clinic that complements the capacity measures obtained in consultation. However, the usage of smartphones is very heterogeneous amongst the population, particularly regarding the position on the body (e.g., pockets, hand, shoulder bag).

The relative reliability of the gait metrics was estimated by comparing trials’ average values obtained with the different smartphone locations. This relative reliability was excellent with all ICC values greater than 0.85, supporting the high consistency of the presented method. This observation is in line with the already published evidence showing that larger trials improve reliability [6]. Absolute reliability was calculated to assess the variability in the gait-related measurements upon repeated testing. The high relative reliability associated with the low absolute reliability is supportive of the use of the presented method as a reliable clinical assessment. Additionally, SEM was in general smaller than the intra-subject variability, estimated as the standard deviation between trials, pointing to a strong indication that small differences in gait performance are captured. MDCs for all temporal metrics were observed in the same orders of magnitude for the differences obtained when comparing PD patients and healthy controls and between PD patients in the ON and OFF conditions [28,29,30]. This is expected since most temporal metrics do not show statistically significant differences between PD and control or between ON and OFF conditions. On the other hand, stride length and velocity showed larger differences between PD patients and healthy paired controls [28,29,30]. In these cases, the MDC values were very small for all smartphone locations (within the same range as previously observed [5]). This is a very strong indication that this approach is able to capture clinically relevant differences in both stride length and velocity. This emphasizes the relevance and applicability of the presented method for informing clinical decision-making.

The concurrent validity was evaluated using a Bland-Altman analysis with data collected in a previous study with an optoelectronic criterion and a smartphone located in a shorts pocket. The biases and LoA were slightly higher than those observed in the previous work [23]. Stride length and stride velocity biases were much smaller (2–3 orders of magnitude) of those obtained when comparing PD patients and control subjects and between ON and OFF states of PD patients [28,29,30]. Regarding the temporal metrics, the observed uncertainties were within the same range as those in these previous works that did not, however, show statistically significant differences.

A second Bland-Altman analysis was performed to evaluate the concurrent validity of Belt, Hand, Pocket and Bag against the Pants pocket location. All biases were three orders of magnitude lower than the average value, which means that on average, all locations are measuring the same. The LoAs were at least one order of magnitude lower than the average value, meaning that the different locations measured with the same precision. Moreover, all LoAs were much smaller than the 30% cut-off suggested in previous studies [31,32]. This result is a consequence of the very strict protocol for detecting and filtering gait events that on the one hand discards a very significant amount of data (Table 1) but on the other hand ensures that the results are consistent between different locations. Additionally, in an ecological environment, very significant variations from a standard gait event are expected: sudden changes of direction, hesitations, people talking and gesturing with the phones in their hand, etc. In this study, some of these situations were mimicked, which explains the amount of data that was labelled as outlier data. Moreover, as expected, the percentage of outliers strongly depends on the smartphone position, with the Hand being the position where more data were discarded from given the higher observed noise. The exclusion of a significant amount of data is, however, not problematic in an ecological context where it is expected that many gait events are collected every day.

The relative values of LoA for the temporal metrics were consistent with its magnitude, i.e., the largest LoAs were observed for stride time, the lowest for swing and intermediate for stance. Unexpectedly, the smaller LoA values were observed for stride length regardless of device position. This is also associated with the outlier exclusion procedure. The stride length model that was used depends on the variance of the vertical component of the acceleration signal and on the stride frequency. Since an outlier removal step was used, most gait cycles have a standard pattern. This results in a consistent result for the stride length.

Being based on a single algorithm for all smartphone locations, this method eliminates the necessity of a priori knowledge or of a location detection method. Agnosticism to location also allows for normal smartphone usage, preventing high dropout rates and increasing ecological validity [8].

The presented results are in agreement with the high reliability reported in previous studies (within the 10% range) of healthy subjects and in controlled environments for the bag and belt locations [14]. Furthermore, as previously mentioned, the strict outlier exclusion procedure meant it was possible to improve on the results for the hand and pocket locations. This work also further extends previously obtained reliability results on the gait of PD patients [13,17,18].

## 5. Conclusions

This paper describes a method for obtaining spatiotemporal gait metrics in an approximately ecological environment with high reliability and validity. The key to achieving these results was to develop a very thorough protocol that learns and filters excess noise and, consequently, only selects standard gait cycles. With this protocol, stride time and cadence, stance time, swing time, stride length and velocity can be accurately quantified in PD patients during real-world walking independent of the smartphone’s location on the body. The presented results support that this smartphone-based methodology capture clinically relevant gait metrics in an ecological context, which constitutes an important step towards patient-centred remote care.

## 6. Patents

Ricardo Matias (Kinetikos Driven Solutions S.A.) is the sole inventor of pending Portuguese provisional patent application No. 117597, filed by applicant Kinetikos Driven Solutions S.A., entitled “System and Method for Unsupervised Monitoring of Mobility Related Disorders”.

## Figures and Tables

**Figure 1 jpm-12-00826-f001:**
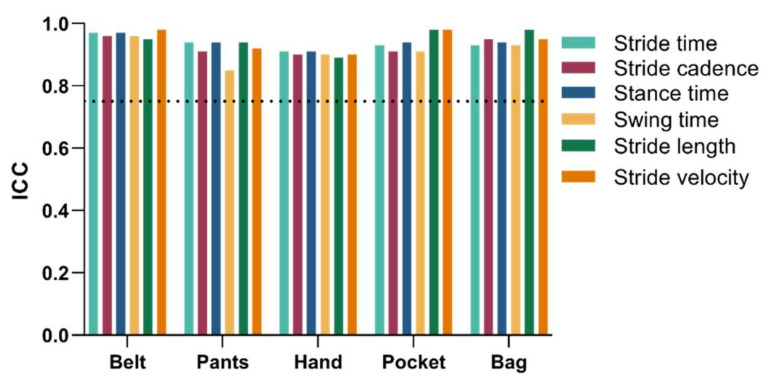
ICC values for relative reliability. The dotted line indicates the 0.75 threshold for excellent agreement.

**Figure 2 jpm-12-00826-f002:**
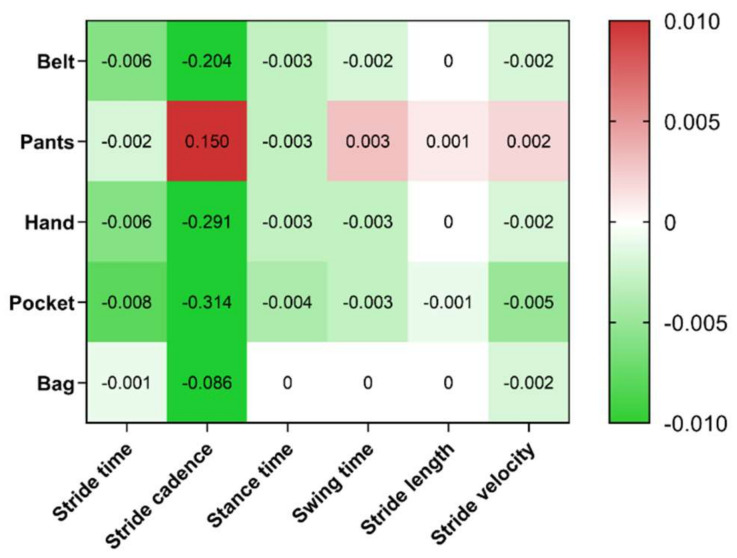
The differences between SEM and the intra-subject variability for each metric and smartphone location.

**Figure 3 jpm-12-00826-f003:**
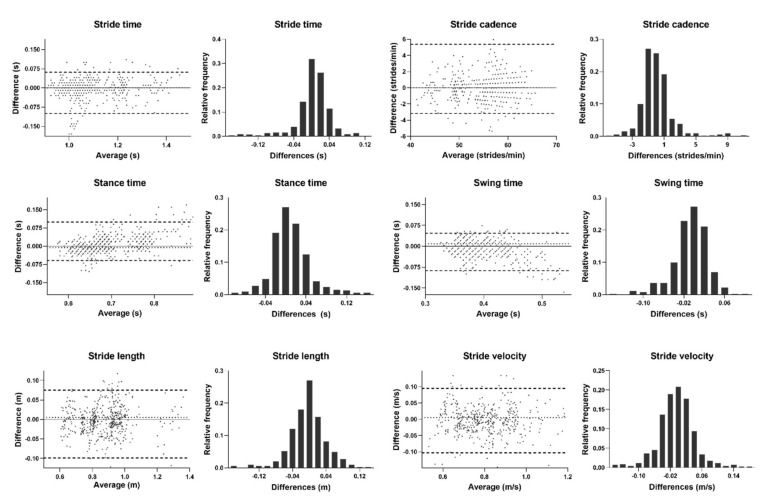
Bland-Altman analysis of the differences between gait metrics obtained in a pants pocket and the optoelectronic criterion versus the median of the two measurements. Distribution plots of differences between methods are presented for each metric. Dashed lines represent bias (median) and the limits of agreement (quantiles 2.5% and 97.5%).

**Figure 4 jpm-12-00826-f004:**
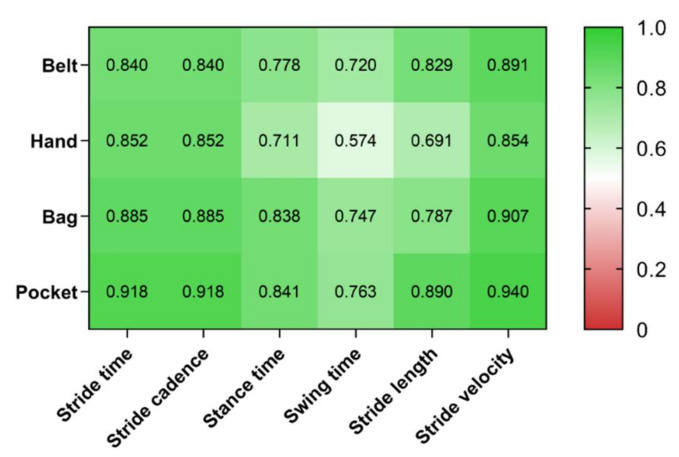
Spearman correlation coefficient (⍴) with Pants location for all other locations and for all gait metrics (*p* < *0*.001 for all combinations).

**Table 1 jpm-12-00826-t001:** The data that remained after each of the two main steps in the gait analysis protocol. Using Belt as an example, 96.6% of the initial data remained after event detection, and 73.5% remained after outlier exclusion.

Location	Event Detection (%)	Outlier Exclusion (%)
All	96.3	61.5
Belt	96.6	73.5
Pants	97.3	51.5
Hand	96.6	39.5
Pocket	93.8	68.0
Bag	97.4	75.2

**Table 2 jpm-12-00826-t002:** Summary of the demographic characteristics and clinical data of the sample of 20 PD patients.

Demographic and Clinical Data	
Age (mean, SD)	64.3 ± 10.6
Male sex (% (*n*))	55 (11)
Time since diagnosis (mean, SD)	7.65 ± 5.6
Presence of motor fluctuation (% (*n*))	50 (10)
Presence of dyskinesias (% (*n*))	40 (8)
Presence of freezing (% (*n*))	50 (10)
MDS-UPDRS I (range 0–52)	10.2 ± 7.9
MDS-UPDRS II (range 0–52)	10.0 ± 6.7
MDS-UPDRS III (range, 0–132)	25.4 ± 15.9
MDS-UPDRS IV (range 0–24)	3.2 ± 3.3
MDS-UPDRS Total (range 0–260)	48.7 ± 26.5
Hoehn and Yahr stage (range 1–5)	2 ± 0.6
TUG (s)	9.4 ± 3.0
CGI–S (range 0–7)	3.1 ± 0.9
PGI–S (range 0–7)	3.4 ± 1.0
PDQ-39 (Median (Min, Max); range 0–156)	33.5 (3, 80)

**Table 3 jpm-12-00826-t003:** ICC values for relative reliability.

Location	Stride Time (s)	Stride Cadence (Strides/min)	Stance Time (s)	Swing Time (s)	Stride Length (m)	Stride Velocity (m/s)
Belt	0.97	0.96	0.97	0.96	0.95	0.98
Pants	0.94	0.91	0.94	0.85	0.94	0.92
Hand	0.91	0.90	0.91	0.90	0.89	0.90
Pocket	0.93	0.91	0.94	0.91	0.98	0.98
Bag	0.93	0.95	0.94	0.93	0.98	0.95

**Table 4 jpm-12-00826-t004:** Intra-subject variability, SEM and MDC for each metric and smartphone location.

Metric	Location	Intra-Subject Variability	SEM	MDC
Stride time (s)	Belt	0.025	0.019	0.044
	Pants	0.032	0.030	0.070
	Hand	0.038	0.032	0.075
	Pocket	0.026	0.018	0.043
	Bag	0.019	0.018	0.041
Stride cadence (strides/min)	Belt	1.188	0.984	2.297
	Pants	1.847	1.997	4.660
	Hand	1.844	1.553	3.625
	Pocket	1.321	1.007	2.349
	Bag	0.885	0.799	1.865
Stance time (s)	Belt	0.015	0.012	0.027
	Pants	0.020	0.017	0.039
	Hand	0.023	0.020	0.047
	Pocket	0.014	0.010	0.024
	Bag	0.010	0.010	0.023
Swing time (s)	Belt	0.010	0.008	0.018
	Pants	0.017	0.020	0.047
	Hand	0.015	0.012	0.029
	Pocket	0.011	0.008	0.019
	Bag	0.006	0.006	0.015
Stride length (m)	Belt	0.008	0.008	0.018
	Pants	0.008	0.009	0.022
	Hand	0.014	0.014	0.034
	Pocket	0.009	0.008	0.019
	Bag	0.004	0.004	0.010
Stride velocity (m/s)	Belt	0.014	0.012	0.027
	Pants	0.027	0.029	0.067
	Hand	0.031	0.029	0.067
	Pocket	0.017	0.012	0.028
	Bag	0.013	0.011	0.026

**Table 5 jpm-12-00826-t005:** Agreement between gait parameters measured in the pants pocket and the optoelectronic criterion. Bias and LoA from the Bland-Altman analysis are shown.

Metric	Bias	LoA (Lower)	LoA (Upper)	LoA (%)
Stride time (s)	0.000	−0.100	0.061	7.6
Stride cadence (strides/min)	0.000	−3.169	5.357	7.5
Stance time (s)	0.006	−0.059	0.099	11.9
Swing time (s)	−0.009	−0.088	0.047	17.4
Stride length (m)	−0.005	−0.099	0.075	10.0
Stride velocity (m/s)	−0.005	−0.103	0.095	12.7

**Table 6 jpm-12-00826-t006:** Agreement between gait parameters measured in the pants pocket and other smartphone locations. Bias and LoA from the Bland-Altman analysis are shown.

Metric	Location	Bias	LoA (Lower)	LoA (Upper)	LoA (%)
Stride time (s)	Belt	0.002	−0.104	0.108	9.6
	Hand	−0.006	−0.111	0.106	9.7
	Pocket	0.000	−0.107	0.082	8.8
	Bag	−0.003	−0.085	0.060	6.5
Stride cadence (strides/min)	Belt	0.057	−5.357	5.288	9.8
	Hand	−0.287	−5.100	4.801	9.3
	Pocket	0.000	−5.381	4.557	9.0
	Bag	−0.166	−4.272	2.888	6.7
Stance time (s)	Belt	0.000	−0.077	0.074	10.9
	Hand	0.004	−0.094	0.096	13.5
	Pocket	0.000	−0.057	0.074	9.6
	Bag	0.005	−0.053	0.068	8.5
Swing time (s)	Belt	−0.001	−0.054	0.051	12.9
	Hand	−0.002	−0.067	0.077	17.5
	Pocket	0.000	−0.045	0.054	12.4
	Bag	−0.001	−0.047	0.050	11.8
Stride length (m)	Belt	0.005	−0.030	0.045	4.4
	Hand	−0.001	−0.082	0.044	7.3
	Pocket	−0.003	−0.049	0.038	5.1
	Bag	0.016	−0.034	0.046	4.7
Stride velocity (m/s)	Belt	0.006	−0.062	0.082	9.3
	Hand	−0.007	−0.110	0.070	11.9
	Pocket	−0.004	−0.072	0.062	8.4
	Bag	0.011	−0.045	0.060	7.0

## Data Availability

The data presented in this study are available on request from the corresponding author.

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
