# Peer review of "Smartphone-Based Body Location-Independent Functional Mobility Analysis in Patients with Parkinson’s Disease: A Step towards Precise Medicine"

_jpm, 2022, doi:10.3390/jpm12050826_

Round 1

Reviewer 1 Report

I believe that the authors could add photos of the places where the smartphones were placed, to improve the understanding of the readers.

Author Response

We thank the reviewer for the comment. A picture representing the smartphones location was added.

Reviewer 2 Report

The work is devoted to the analysis of the control of patients with Parkinson's disease, who are on the move, using a smartphone located in various places of the body. A specific evaluation protocol has been developed for the selection of standard gait cycles. The five most common different positions of the phone are considered. Statistical processing of the results was carried out. First, the discarded data were screened out, then the remaining results were statistically processed. The conclusions drawn indicate a high reliability of the results of the assessment of walking.

There are small questions.

  1. Where did the formula on line 156 come from?
  2. In table 1, it is desirable to add units of measurement (percentages) to the headings.
  3. Can you explain the difference in the results in Table 1 between different cases of wearing a smartphone.
  4. Do not define abbreviations several times: SEM and others.
  5. The graphs in Figure 3 are very hard to see.
  6. The question is more theoretical. Can the results of assessing the walking of patients be improved if values from different places on the human body are used.

Author Response

  1. We thank the reviewer for the comment. Indeed, the references for the original works were missing. They were now added.
  2. The units were added to the headings
  3. We agree with the reviewer that this was missing in the original manuscript. A few lines commenting these results were added to the discussion.
  4. We thank the reviewer for the comment and review the definitions of some abbreviations.
  5. The resolution of Figure 3 was improved.
  6. Theoretically, every algorithm has room for improvement. This study showed that, regardless of the smartphone position with respect to the body, accurate gait-related parameters can be computed. Our previous research also showed us that adding more sensors does not lead us to more accuracy for classical spatiotemporal parameters (10.1186/s12883-021-02354-x). Having said this, by reducing the number of sensors to one and making it position-independent we are paving the way for ecological gait measurements with reduced user burden.

Reviewer 3 Report

Manuscript ID: jpm-1716784

Title: Smartphone-based body location-independent functional mobility analysis in patients with Parkinson’s Disease: a step towards precise medicine

Recommendation: Revise

Brief summary

This study analyses the effect of the different positioning (pants pocket, belt, hand, shirt pocket and shoulder bag) of smartphone in the assessment of the gait of patients affected by Parkinson disease.

Broad comments

The topic is relevant, since assessing the walking ability of PD patients in a minimally invasive way can be of real benefit. Moreover, wearable sensors are continuously spreading.

The manuscript is written quite well and the English is generally fluent, even if some typos are present. Hence, thoroughly re-reading the papers could help authors to correct eventual typos and also to improve the use of punctuation in order to improve the manuscript readability.

Moreover, the article is quite well contextualized in the literature background, even in some references in the domain of JPM could be added.

Some suggestions are provided in the next comments, which may help the authors in improving the quality of this paper.

Specific comments

Abstract: it is not clear how the authors evaluated the reliability and validity of data (reference device? Reference positioning?). Please clarify this aspect within the abstract.

Line 20: please always insert a spacing between a number and its related measurement unit.

Line 23: please spell out the acronym ICC when used for the first time.

Lines 114-117: it could be useful to add a scheme reporting the different steps of the test.

Lines 120-123: it could be useful to insert a figure reporting the experimental configuration.

Line 124: it could be useful to insert a flow chart reporting the pipeline of data processing. Moreover, the authors should better explain which the reference is for the comparison of results from smartphones positioned in different locations. This should be clarified from the very early of the data analysis description.

Lines 125-127: the authors should insert the technical specifications of the sensors employed in the study, such as sampling frequency, resolution, etc.

Lines 150-154: perhaps it would be better to insert a bullet point list to describe the parameters.

Line 156: please evaluate if all the reported decimal digits are significant.

Table 3: the measurement units should be reported.

Figure 3: the figure resolution should be improved. Moreover, please always insert the measurement units on the axes.

Lines 276-283: the authors should try to motivate the differences with respect to the reported previous study.

Line 296: the authors could report how many data were classified as outliers, to ease readability and comprehension of the work.

Line 300: the authors should quantify the “very small” LoA, for ease of readability.

Line 309: the “high reliability” should be quantified, both for previous study and for the present one.
